# How Does CBG Administration Affect Sphingolipid Deposition in the Liver of Insulin-Resistant Rats?

**DOI:** 10.3390/nu15204350

**Published:** 2023-10-12

**Authors:** Wiktor Bzdęga, Piotr Franciszek Kurzyna, Ewa Harasim-Symbor, Adam Hołownia, Adrian Chabowski, Karolina Konstantynowicz-Nowicka

**Affiliations:** 1Department of Physiology, Medical University of Bialystok, 15-089 Bialystok, Poland; wbzdega@gmail.com (W.B.); pf.kurzyna@gmail.com (P.F.K.); ewa.harasim-symbor@umb.edu.pl (E.H.-S.); adrian@umb.edu.pl (A.C.); 2Department of Pharmacology, Medical University of Bialystok, 15-089 Bialystok, Poland; adam.holownia@umb.edu.pl

**Keywords:** cannabigerol, ceramide, sphingolipid metabolism, insulin resistance, metabolic disorders, liver

## Abstract

Background: Cannabigerol (CBG), a non-psychotropic phytocannabinoid found in *Cannabis sativa* plants, has been the focus of recent studies due to its potential therapeutic properties. We proposed that by focusing on sphingolipid metabolism, which plays a critical role in insulin signaling and the development of insulin resistance, CBG may provide a novel therapeutic approach for metabolic disorders, particularly insulin resistance. Methods: In a rat model of insulin resistance induced by a high-fat, high-sucrose diet (HFHS), we aimed to elucidate the effect of intragastrically administered CBG on hepatic sphingolipid deposition and metabolism. Moreover, we also elucidated the expression of sphingolipid transporters and changes in the sphingolipid concentration in the plasma. Results: The results, surprisingly, showed a lack of changes in de novo ceramide synthesis pathway enzymes and significant enhancement in the expression of enzymes involved in ceramide catabolism, which was confirmed by changes in hepatic sphingomyelin, sphinganine, sphingosine-1-phosphate, and sphinganine-1-phosphate concentrations. Conclusions: The results suggest that CBG treatment may modulate sphingolipid metabolism in the liver and plasma, potentially protecting the liver against the development of metabolic disorders such as insulin resistance.

## 1. Introduction

*Cannabis sativa* plant (marijuana) contains a wide spectrum of phytochemicals including over 100 cannabinoids, terpenoids, sterols, and flavonoids [1]. One of them is cannabigerol (CBG), which is a non-psychotropic phytocannabinoid. It belongs to phenolic terpene, which naturally occurs in the Cannabis sativa L. plant, mainly in female inflorescence. CBG content varies on the plant species—e.g., there are particular industrial hemp varieties abundant in CBG [2,3].

Studies indicate that CBG involves a multidirectional pharmacological action throughout several targets, inter alia, transient receptor potential (TRP) channels, cyclooxygenase (COX-1 and COX-2), the serotonin 1A receptor (5-HT1A), and alpha-2 receptors, and likewise the transcriptional activity of peroxisome proliferator-activated nuclear receptor γ (PPARγ), nuclear factor kb (NF-kb), and nuclear factor of kappa light polypeptide gene enhancer in B-cells inhibitor alpha (IκB-α) and enzymes (superoxide dismutase 1 [SOD-1], and inducible nitric oxide synthase [iNOS]) [2,4,5].

In recent years, researchers have paid great attention to the components found in Cannabis, due to their pleiotropic effects on the human body. Relatively few studies have sought to investigate the pharmacological actions of CBG; however, the available preclinical studies have demonstrated that CBG possesses strong antioxidant and anti-inflammatory activities [5]. Its effectiveness has been demonstrated in the treatment of gastrointestinal disorders (inflammatory bowel disease) [6], metabolic syndrome [7], and neurological disorders (Multiple Sclerosis, Parkinson’s disease) [7,8]. What is more, it has anti-anxiety, anti-nausea, and anti-tumoral activity, and thus could also represent a novel adjuvant therapy in oncology [4].

We suspect that cannabigerol may serve as a novel, complementary pharmacologic treatment of insulin resistance (IR). IR is defined as an impaired action of insulin in tissues, through the disruption of different molecular pathways in insulin-dependent target tissues [9,10]. The problem is even more urgent, since it accompanies numerous metabolic pathologies which are considered to be the 21st-century global epidemics, such as obesity, diabetes, metabolic-associated fatty liver disease (MAFLD), and cardiovascular diseases. IR is one of the most common metabolic abnormalities—the worldwide presence of IR among adults varies across countries, but ranges from 15.5 to 46.5%, with higher prevalence in developed countries with a predominance of the Western dietary pattern [11,12].

It is known that the high availability of energy substances (intensified fatty acid influx) exceeding the oxidative capacity of the cell leads to the excessive accumulation of lipids in tissues not adapted to it [13,14]. These lipids, undergoing esterification, lead to the excessive accumulation of bioactive lipid derivatives, such as sphingolipids, which interfere with many signaling pathways, e.g., insulin signal transduction, leading to the development of IR [15]. Sphingolipids are an active class of lipids that constitute components of cell membranes and play a crucial role in the numerous cell metabolism processes as signaling molecules [15]. Their action on cells varies; however, their excessive accumulation has been associated with an overall exacerbation of IR [16,17]. Although some effects of CBG are partially established, its influence on sphingolipid metabolism, which may affect hepatic insulin resistance, remains uncharted. Research on CBG deserves to be advanced to fill an important gap in knowledge to develop novel therapeutic approaches for metabolic disorders. Thus, our study aims to elucidate the effect of cannabigerol on liver sphingolipid deposition and metabolism in the rat model of insulin resistance induced by a high-fat high-sucrose diet (HFHS).

## 2. Materials and Methods

### 2.1. Animals and Experimental Protocol

The experiment took place over a span of six weeks and involved male Wistar rats initially weighing around 70–100 g. Throughout the entire study period, all the animals were kept in standard laboratory animal living conditions: plastic autoclavable cages, an air temperature of 22 ± 2 °C, a 12 h light/dark cycle, and unrestricted access to either water or sucrose solution, along with a selected rodent diet. Following a one-week adjustment period to their new surroundings, the rats were arbitrarily split into four groups, each consisting of 10 rats. The experimental groups (Figure 1) were as follows: (1) control group—rats were fed standard rodent chow (Labofeed B standard, Morawski, Kcynia, Poland) containing 2.74 kcal/g (60% of the calories from carbohydrates, 25% from protein, and 8% from fat) and water to drink; (2) CBG group—rats fed the above-described standard diet, water, and receiving cannabigerol (TargetMoll, Boston, MA, USA); (3) HFHS group—rats fed high-fat diet (Research Diet, New Brunswick, NY, USA) containing 5.24 kcal/g (60% of the calories from fat, 20% calories from protein, and 20% calories from carbohydrates) and 20% solution of sucrose to drink; (4) HFHS + CBG group—rats fed the above-described high-fat diet and 20% solution of sucrose, and treated simultaneously with CBG. The compound was dissolved in sesame oil and promptly given via intragastric administration at a rate of 30 mg/kg of body weight during the final 14 days of the study. The CBG dosage was adapted in accordance with the available information [18,19]. Administering CBG intragastrically ensured that the rats received the correct dose relative to their body weight. Animals body weight was monitored during the whole experiment. At the conclusion of the sixth week, subsequent to a 12 h overnight fast, rats underwent intraperitoneal administration of pentobarbital (80 mg/kg of body weight) for anesthesia induction. Following anesthesia, venous blood samples were procured from the tail vein and transferred into heparin-containing receptacles. Subsequently, the obtained blood samples underwent centrifugation, resulting in the isolation of plasma for analytical purposes. Concurrently, hepatic specimens were acquired from quiescent subjects and promptly subjected to rapid cryopreservation in liquid nitrogen, employing pre-chilled aluminum tongs. All specimens were preserved at a temperature of −80 °C to facilitate subsequent analyses. Furthermore, a secondary experimental iteration was conducted to assess insulin sensitivity, with the subject rats being randomly partitioned into the identical four groups delineated previously, encompassing six rats within each experimental cohort. The study was approved by the Ethical Committee for Animal Testing in Olsztyn, Poland (permission no. 19/2022).

### 2.2. Insulin Tolerance Test

To determine CBG influence on the insulin sensitivity the Insulin Tolerance Test (ITT) was conducted on animals during second set of experiments, and n = 6 rats were used. The animals were fasted overnight for 12 h and injected intraperitoneally with human insulin (Gensulin) in a dose of 0.75 IU/kg of body weight. The blood was collected from tail vein before insulin injection (0) and 15, 30, 45, 60, 90, and 120 min post injection. Blood glucose concentration was measured using glucose analyzer and test strips (Accu-Chek). Data obtained from glucose levels were plotted as graphs, and AUC was calculated.

### 2.3. Determination of the Plasma and Tissue Sphingolipid Concentration

The concentration of sphingosine, sphingosine-1-phosphate, sphinganine, sphinganine-1-phosphate, and ceramide was quantified employing the high-performance liquid chromatography (HPLC) technique. This procedure was described in detail by Baranowski et al. [20]. The liver was homogenized, and lipids from the tissue and plasma samples were extracted into the chloroform phase. The obtained lipid extracts aliquots were transferred to a fresh tube containing 40 pmol of N-palmitoyl-D-erythro-sphingosine (C17 base) as an internal standard. Next, the alkaline water was added to the samples to form deacylated ceramide. Released free sphingosine and sphinganine from ceramide were converted to their o-phthalaldehyde derivatives and analyzed using HPLC system (Varian ProStar, Agilent Technologies) with a fluorescence detector and C18 reversed-phase column (Varian Inc. OmniSpher 5, 4.6 × 150 mm, Lake Forest, CA, USA). The sphingolipids concentration was presented in picomoles per milligram of tissue or picomoles per milliliter of plasma.

### 2.4. Determination of the Plasma and Tissue Sphingomyelin Concentration

To concisely summarize, lipid extraction from both liver and plasma samples was accomplished utilizing a chloroform/methanol solution with a volumetric ratio of 2:1, following the established protocol described by Folch et al. [21]. Subsequently, the resulting extracts were applied onto glass chromatographic plates coated with silica gel (Silica Plate 60, 0.25 mm; Merck, Darmstadt, Germany) and subjected to separation to isolate the sphingomyelin fraction, employing Thin-Layer Chromatography (TLC) methodology. The eluents obtained were subjected to trans-methylation in a 14% boron trifluoride-methanol solution, followed by dissolution in hexane. This was followed by the utilization of gas–liquid chromatography (GLC), performed using a Hewlett-Packard 5890 Series II gas chromatograph equipped with a flame ionization detector and Hewlett-Packard-INNOWax capillary column (Agilent Technologies, Santa Clara, CA, USA), for the identification and quantification of individual fatty acid methyl esters based on the retention times of established standards.

### 2.5. Determination of Glycogen Concentration

Quantitative determination of glycogen level in the hepatic tissue homogenates was evaluated using a commercially available kit (Glycogen Assay Kit II; cat no.: ab169558, Abcam, Cambridge, UK). The above analysis was based on the colorimetric method and was performed according to the manufacturer’s instructions. The absorbance of the color product was measured at the wavelength of 450 nm using Synergy H1™, BioTek Instruments, Winooski, VT, USA. The final concentration of glycogen was expressed in micrograms per microliter (µg/µL).

### 2.6. Western Blotting

A routine Western blotting procedure was used to assess protein expression, as was described in detail previously [16]. In brief, before the immunoblotting procedure, the liver was homogenized in radioimmunoprecipitation assay (RIPA) buffer at 4 °C with the addition of protease and phosphatase inhibitors. After centrifugation, total protein concentration was determined with bicinchoninic acid (BCA) assay. All the proteins (30 μg) were electrophoretically separated using 10% Criterion TGX Stain-Free Precast Gel (Bio Rad, Poland) and transferred onto nitrocellulose or polyvinylidene fluoride (PVDF) membranes. Then, the membranes were blocked with 5% BSA or 5% non-fat dry milk and immunoblotted with primary antibodies of interest from sphingolipid pathway: SPTLC1, SPTLC2 (Santa Cruz Biotechnology, Dallas, TX, USA), SPHK1, SPHK2 (Sigma Aldrich, Darmstadt, Germany), CerS2 (Santa Cruz Biotechnology, Dallas, TX, USA), CerS4 (Sigma Aldrich, Darmstadt, GER), CerS5 (Thermo Fisher Scientific, Waltham, MA, USA), CerS6 (Abnova, Taipei, Taiwan), ASAH1 (Abcam, Cambridge, UK), ASAH2 (Santa Cruz Biotechnology, Dallas, TX, USA), ASAH3 (Thermo Fisher Scientific, Waltham, MA, USA), N-SMase, Alk-SMase (Santa Cruz Biotechnology, Dallas, TX, USA), SPSN2, SGPL1, S1PR2 (Thermo Fisher Scientific, Waltham, MA, USA), S1PR3, CERT (Abcam, Cambridge, UK), ABCA1(Thermo Fisher Scientific, Waltham, MA, USA), insulin signaling pathway: Akt (Cell Signaling Technology Inc., Danvers, MA, USA), pAkt Ser 472, 473, 474 (Abcam, Cambridge, UK), pAkt Thr 308, 309, 305 (Santa Cruz Biotechnology, Dallas, TX, USA), GSK-3α/β (Invitrogen, Waltham, MA, USA), pGSK-3β Ser 9, pGSK-3α Ser 21 (Cell Signaling Technology Inc., Danvers, MA, USA), pGSK-3α Tyr 279, pGSK-3β Thr 216 (Santa Cruz Biotechnology, Dallas, TX, USA) as well as lipid and glucose metabolism: FAS (Cell Signaling Technology Inc., Danvers, MA, USA), SREBP-1c, ACC2 (Santa Cruz Biotechnology, Dallas, TX, USA), phosphoACC2 Ser 9 (Cell Signaling Technology Inc., Danvers, MA, USA), and PDH (Abcam, Cambridge, UK).

Subsequently, the blots underwent incubation with secondary antibodies conjugated with horseradish peroxidase. Following this step, signal visualization was achieved through the utilization of a chemiluminescence substrate (Clarity Western ECL Substrate; Bio-Rad, Hercules, CA, USA), and the resultant signals were subjected to densitometric quantification utilizing a ChemiDoc visualization system (BioRad, Hercules, CA, USA). The overall expression levels of the target proteins were quantitatively assessed using stain-free gels and adopting the total protein normalization technique [22], facilitated by ImageLab software Version 6.0.1 (BioRad, Hercules, CA, USA). The superimposition of the total protein and the protein of interest images enabled the generation of normalized protein expression values, with the control group serving as the reference, set at 100%. Appendix A contains representative images of uncropped, untouched, full original Western blot images in the same protein order as presented in the Section 3. The samples on every blot are loaded in the order as follows: control, CBG, HFHS, HFHS + CBG (n = 6 for each group).

### 2.7. Histology Analysis

Samples of the liver (the same fragment of the lobe from each rat) were collected for histologic studies. They were fixed in 10% formalin and processed routinely for embedding in paraffin. Sections were cut at pieces 4 µm in thickness, and stained with hematoxylin–eosin (H + E). The sections then were briefly rinsed in 50 per cent ethanol followed by tap water, and the nuclei were stained with Mayer’a hematoxylin for 2 min. The sections were rinsed in tap water using the customary procedure and mounted in glycerine jelly. The results of staining were submitted for evaluation in Olympus BX41 microscope with an Olympus DP12 camera under a magnification of 200× (20× lens and 10× eyepiece) with an Olympus CellSens Imaging Software Version 3.1.

### 2.8. Statistical Analysis

The data from the experiment are presented as the mean ± SD or percentage of the control group. The normality of data distribution and the homogeneity of variance were assessed employing the Shapiro–Wilk and Bartlett’s tests, respectively. To determine statistical distinctions, a two-way analysis of variance (ANOVA) was conducted, succeeded by a suitable post hoc test (Tukey or pairwise Student’s *t*-test). A significance level of *p* < 0.05 was established as the threshold for statistical significance in all instances. The statistical analyses were executed using GraphPad Prism Version 8 software.

## 3. Results

### 3.1. The Effect of CBG Treatment on Sphingolipid Concentration in the Liver

The concentration of SFO in the liver was significantly reduced in the group treated with CBG compared to the control group (CBG: −28.3%, *p* < 0.05) (Figure 2A). Moreover, CER concentration in the liver was also markedly diminished in the same group (CBG: −18.3%, *p* < 0.05) (Figure 2E). Regarding the liver’s S1P content, we observed elevated accumulation in CBG, HFHS, and HFHS + CBG groups compared to the control group (CBG: +32.6%, HFHS: +58.6%, HFHS + CBG: +142.8%, *p* < 0.05) (Figure 2C), as well as in the HFHS + CBG group compared to HFHS group (HFHS + CBG: +53.1%, *p* < 0.05) (Figure 2C). The concentration of SFA was increased in the liver of rats treated solely with CBG and decreased in the group fed HFHS in comparison with the control group (CBG: +76.8%, HFHS: −83.8%, *p* < 0.05) (Figure 2B). There was also a significant increase in SFA accumulation in HFHS + CBG compared to the HFHS group (HFHS + CBG: +471.0%, *p* < 0.05) (Figure 2B). Likewise, in groups solely treated with CBG or fed HFHS, SFA1P concentration in the liver was elevated compared to the control group (CBG: +85.8%, HFHS: +52.1%, *p* < 0.05) (Figure 2D), while in group simultaneously fed HFHS and treated with CBG, in comparison with the HFHS group its concentration was diminished (HFHS + CBG: −41.5%, *p* < 0.05) (Figure 2D). We observed lowered accumulation of liver’s SPH in rats fed HFHS compared to the control group (HFHS: −25.7%, *p* < 0.05) (Figure 2F) and an increased accumulation of this lipid in HFHS treated with CBG in comparison with HFHS group (HFHS + CBG: +20.9%, *p* < 0.05) (Figure 2F).

### 3.2. The Effect of CBG Treatment on Sphingolipid Concentration in the Plasma

The concentration of SFA was considerably increased in the group treated only with CBG in relation to the control (CBG: +25.7%, *p* < 0.05). In the groups fed a HFHS diet (with or without the addition of CBG), we observed markedly elevated concentration of SFA (HFHS: +101.6%, HFHS + CBG: +140.0%, *p* < 0.05) (Figure 3B) and CER in the plasma (HFHS: +51.3%, HFHS + CBG: +98.0%, *p* < 0.05) (Figure 3E) compared to the control group. Moreover, CER accumulation was also significantly increased in comparison with the HFHS group (HFHS + CBG: +30.9%, *p* < 0.05) (Figure 3E). In the groups simultaneously treated with CBG and fed a HFHS diet we observed substantially reduced concentration of S1P in plasma compared to the control group (HFHS + CBG: −15.7%, *p* < 0.05) (Figure 3C) and HFHS group (HFHS + CBG: −25.7%, *p* < 0.05) (Figure 3C). Similarly, we noticed the diminished accumulation of SFA1P in plasma in the HFHS + CBG group in comparison to the control (HFHS + CBG: −31.3%, *p* < 0.05) (Figure 3E) and HFHS groups (HFHS + CBG: −27.8%, *p* < 0.05) (Figure 3E). Moreover, there was an elevation in SFA1P in the group treated with CBG solely compared to the control group (CBG: +39.5%, *p* < 0.05) (Figure 3E). We did not find any substantial changes in the concentrations of SFO and SPH in the plasma (*p* < 0.05) (Figure 3A,F).

### 3.3. The Effect of CBG Treatment on the Expression of Enzymes Specific to the De Novo Ceramide Synthesis Pathway

The only important change which was found was in the group fed HFHS, where we observed a significant decrease in both enzymes’ expression, namely SPTLC1 (HFHS: −32.4%, *p* < 0.05) (Figure 4A) and SPTLC2 (HFHS: −20.4%, *p* < 0.05) (Figure 4B), in comparison with the control group.

### 3.4. The Effect of CBG Treatment on the Expression of Enzymes That Share Salvage and De Novo Ceramide Synthesis Pathways 

We did not observe any significant changes in CerS2 and CerS4 expression (*p* < 0.05) (Figure 5A,B). In contrast, we noticed important changes in CerS5 and CerS6 expression. The expression of CerS5 was substantially decreased in the group treated with CBG alone, as well as in the HFHS diet with the addition of CBG group in comparison with the control (CBG: −40.3%, HFHS + CBG: −34.8%, *p* < 0.05) (Figure 5C). Moreover, rats fed a HFHS diet and treated with CBG simultaneously exposed markedly lowered expression of CerS5 compared to the HFHS group (HFHS + CBG: −38.4%, *p* < 0.05) (Figure 5C). Interestingly, the CerS6 expression decreased in the HFHS group compared to the control (HFHS: −33.3%, *p* < 0.05) (Figure 5D), and significantly increased in the HFHS + CBG in relation to the HFHS group (HFHS + CBG: +93.1%, *p* < 0.05) (Figure 5D).

### 3.5. The Effect of CBG Treatment on the Expression of Enzymes from the Ceramide Catabolic Pathway

In the CBG and HFHS groups, a substantial increase in ASAH2 expression was found in comparison with the control group (CBG: +33.0%, HFHS: +49.3%, *p* < 0.05) (Figure 6B). Moreover, we noticed a significant rise in ASAH3 expression in the groups fed a HFHS diet with or without CBG compared to the control group (HFHS: +57.6%, HFHS + CBG: +157.4%, *p* < 0.05) (Figure 6C) and HFHS with CBG in relation to the HFHS group (HFHS + CBG: +63.4%, *p* < 0.05) (Figure 6C). Moreover, we observed similar changes in SPHK1 expression in comparison with the control (HFHS: +134.7%, HFHS + CBG: +196.1%, *p* < 0.05) (Figure 6D) and HFHS groups (HFHS + CBG: +26.1%, *p* < 0.05) (Figure 6D). In contrast, SPHK2 expression was markedly decreased in the group treated simultaneously with HFHS and CBG compared to the control group (HFHS + CBG: −24.1%, *p* < 0.05) (Figure 6E). No significant changes in ASAH1 expression were found (*p* < 0.05) (Figure 6A).

### 3.6. The Effect of CBG Treatment on the Expression of Enzymes from Hydrolysis of the Sphingomyelin Pathway and Receptors for S1P

The expression of ALK-SMase was decreased in HFHS + CBG compared to the control group (HFHS + CBG: −20.7%, *p* < 0.05) (Figure 7A), while N-SMase expression was significantly weakened in the rats treated with CBG and fed a HFHS diet simultaneously in comparison with the HFHS group (HFHS + CBG: −25.9%, *p* < 0.05) (Figure 7B). We observed a marked increase in S1PR2 receptor expression in the HFHS compared to the control group (HFHS: +20.4%, *p* < 0.05) and a decrease in the group treated with HFHS + CBG compared to the HFHS group (HFHS + CBG: −23.0%, *p* < 0.05) (Figure 7C). We found an increased expression of S1PR3 in groups fed a HFHS diet with or without the addition of CBG in comparison with the control group (HFHS: +77.3%, HFHS + CBG: +64.7%, *p* < 0.05) (Figure 7D).

### 3.7. The Effect of CBG Treatment on the Expression of S1P-Degrading Enzyme and Sphingolipid Transporting Proteins

In the groups fed a HFHS diet and simultaneously with HFHS and CBG, we observed a substantial decrease in CERT (HFHS: −35.1%, HFHS + CBG: −43.0%, *p* < 0.05) (Figure 8B) and SPSN2 expression (HFHS: −28.9%, HFHS + CBG: −32.1%, *p* < 0.05) (Figure 8C) compared to the respective control group. Similarly, we showed a lowered ABCA1 expression in the HFHS + CBG group in comparison with the HFHS group (HFHS + CBG: −33.9%, *p* < 0.05) (Figure 8D). We did not notice any significant changes in SGPL1 expression (*p* < 0.05) (Figure 8A).

### 3.8. The Effect of CBG Treatment on the Expression of Proteins from the Insulin Signaling Pathway

The administration of CBG to the rats fed a standard and HFHS diet significantly increased the ratio of phosphorylation in Ser 472, 473, and 474 Akt to total Akt compared to the control (CBG: +26.5%, HFHS + CBG: +83.6%, *p* < 0.05) and HFHS group (HFHS + CBG: +161.6%, *p* < 0.05) (Figure 9A). However, the opposite effect was observed in this ratio in the group fed only with a high-fat diet and sucrose (HFHS: −29.8%, *p* < 0.05) (Figure 9A). The ratio of phosphorylation in Thr 308, 309, and 305 Akt to total Akt was considerably decreased in the HFHS group compared to the control (HFHS: −39.4%, *p* < 0.05) and notably raised in the group treated simultaneously with HFHS and CBG in comparison with the HFHS alone group (HFHS + CBG: +36.5%, *p* < 0.05) (Figure 9B). In the group treated with CBG, as well as the group fed a HFHS diet alone, the ratio of phosphorylation in Ser 9 GSK-3β to total GSK-3α/β was notably diminished compared to the control group (CBG: −48.4%, HFHS: −48.0%, *p* < 0.05) (Figure 10A). The ratio of phosphorylation in Ser 21 GSK-3α to total GSK-3α/β compared to the control group was significantly decreased in all the experimental groups (CBG: +26.5%, HFHS + CBG: +83.6%, *p* < 0.05) (Figure 10B). In comparison with the control group, only in rats treated with CBG was the pGSK-3α Tyr 279/GSK-3α/β ratio considerably reduced (CBG: −46.4%, *p* < 0.05) (Figure 11A). Similarly, in the group which received only CBG, the pGSK-3β Tyr 216/GSK-3α/β ratio was markedly reduced (CBG: −50.2%, *p* < 0.05) (Figure 11B). Moreover, this phosphorylation ratio was also decreased in the group fed simultaneously with the HFHS diet and CBG in comparison with the control (HFHS + CBG: −32%, *p* < 0.05) and HFHS groups (HFHS + CBG: −50.1%, *p* < 0.05) (Figure 11B). In the group of animals from HFHS, the pGSK-3β Tyr 216/GSK-3α/β ratio was significantly enhanced compared to the rats fed a standard diet and drinking water (HFHS: +36.4%, *p* < 0.05) (Figure 11B).

### 3.9. The Effect of CBG Treatment on Insulin Sensitivity and Body Weight

To assess whole body insulin sensitivity, the Insulin Tolerance Test was conducted. Fasting glucose levels were higher in the group treated with the HFHS diet than in the control group; however, they did not reach the level of significance. After 15 min of insulin administration, the glucose concentration was considerably increased in the HFHS group compared to the control group (HFHS: +25.3%, *p* < 0.05) (Figure 12D). A similar effect was observed after 30 min in the HFHS group (HFHS: +20.5%, *p* < 0.05) (Figure 12D). Moreover, at this time point a significant decrease in the glucose concentration was observed in the group treated with HFHS + CBG when compared to the HFHS alone group (HFHS + CBG: −15.8%, *p* < 0.05) (Figure 12D). A substantial enhancement in the glucose levels was visible in the HFHS group 45 (HFHS: +14.0%, *p* < 0.05) and 60 (HFHS: +23.5%, *p* < 0.05) minutes after insulin injection (Figure 12D). The area under the curve (AUC) above the glucose concentration was markedly higher in the HFHS group in comparison with the control (HFHS: +17.9%, *p* < 0.05) (Figure 12C). In addition, the glycogen concentration in the liver was substantially increased in both the HFHS and HFHS + CBG groups compared to the control rats receiving a standard diet (HFHS: +56.9%, HFHS + CBG: +19.8%, *p* < 0.05) (Figure 12B). Importantly, we noticed that 2-week CBG application caused a considerable diminishment in the glycogen content under control conditions (CBG: −25.9%, *p* < 0.05), as well as in animals from the HFHS group (HFHS + CBG: −23.6%, *p* < 0.05) compared to the control group and HFHS group, respectively (Figure 12B).

Comparing animals’ body mass, rats fed a high-fat diet and receiving sucrose to drink increased their body weight compared to the rats which ate standard rat chow and drank water (HFHS: +30.0%, *p* < 0.05) (Figure 12A). The animals from the HFHS + CBG group significantly increased their body weight compared to the control group (HFHS + CBG: +19.0%, *p* < 0.05) (Figure 12A); however, their body mass was decreased in comparison with the HFHS group rats (HFHS + CBG: −8.4%, *p* < 0.05) (Figure 12A).

### 3.10. The Effect of CBG Treatment on the Expression of Proteins Associated with Fatty Acids and Glucose Metabolism

Rats fed a standard diet and receiving CBG for 2 weeks exhibited a significantly lower expression of FAS in the liver (CBG: −30.7%, *p* < 0.05) (Figure 13A), whereas in animals on the high-fat diet and sucrose, the expression of this protein was considerably augmented compared to the control rats (HFHS: +94.9%, HFHS + CBG: +35.5%, *p* < 0.05) (Figure 13A). Interestingly, application of CBG to the HFHS group caused a substantial diminishment of FAS expression in comparison with the control rats (HFHS + CBG: −30.5%, *p* < 0.05) (Figure 13A). Similarly, the expression of SREBP-1c precursor in the liver was markedly elevated in both the HFHS and HFHS + CBG groups compared to the control rats (HFHS: +33.2%, HFHS + CBG: +37.3%, *p* < 0.05) (Figure 13B). Simultaneously, the expression of mature SREBP-1c in the liver was substantially reduced in both the HFHS and HFHS + CBG groups in comparison with the control rats (HFHS: −23.7%, HFHS + CBG: −36.6%, *p* < 0.05) (Figure 13C). In addition, CBG administration for 2 weeks caused a further decrease in the expression of mature SREBP-1c in rats on a high-fat diet and sucrose compared to the HFHS alone group (HFHS + CBG: −17.3%, *p* < 0.05) (Figure 13C). We also measured the expression of ACC2 in the liver, which was considerably increased in rats fed a standard diet and receiving CBG for 2 weeks (CBG: +70.2%, *p* < 0.05) compared to the control rats, and diminished in animals of the HFHS + CBG group (HFHS + CBG: −35.3%, *p* < 0.05) compared to the HFHS group alone (Figure 13D). In parallel, a significant increase in the phosphorylation of ACC2 in the liver was observed only in the case of the HFHS group compared to the control rats (HFHS: +70.6%, *p* < 0.05), which was further decreased by 2-week CBG application compared to the HFHS group alone (HFHS + CBG: −36.6%, *p* < 0.05) (Figure 13E). Moreover, the expression of PDH was also measured in the liver homogenates, and it was substantially lower in both the HFHS and HFHS + CBG groups compared to the control rats (HFHS: −36.6%, HFHS + CBG: −50.2%, *p* < 0.05) (Figure 13F). Conversely, the administration of CBG to HFHS rats even lowered the expression of PDH compared to the HFHS group alone; however, the change did not reach the significance level (HFHS: −21.5%, *p* = 0.076) (Figure 13C).

## 4. Discussion

The Cannabis sativa plant has been used therapeutically for thousands of years worldwide [23]. Its inflorescence is abundant in dozens of cannabinoids, and one of the compounds lacking the psychotropic effects typical for THC is CBG [2]. In recent years, the compelling features of cannabinoids have become one of the emerging approaches for developing effective therapy to ameliorate the symptoms of insulin resistance, T2DM, and metabolic syndrome [23,24]. Disruptions in lipid metabolism followed by the onset of insulin resistance represent prevailing characteristics of the aforementioned metabolic disorders. These pose a relevant problem because there is a lack of effective treatment for the aforementioned diseases. The Western pattern diet, abundant in saturated fatty acids and plain sugars, enhances the uptake and deposition of various lipid classes, which may be esterified into lipid fractions that are more biologically active compared with other lipid fractions, namely, a variety of sphingolipids—ceramide, sphingosine, sphinganine, and sphingomyelin [25]. Recently, a growing number of studies have suggested the correlation between dysfunctional sphingolipid metabolism and impaired insulin signaling followed by liver steatosis development [26,27]. Numerous studies have demonstrated the positive effects of phytocannabinoids such as cannabidiol on the liver’s sphingolipid metabolism [28,29]. However, the influence of CBG on the metabolism of sphingolipids in the liver and insulin signaling pathway remains uncharted.

Thus, our present study was undertaken to harness the possible potential of CBG to ameliorate the impaired sphingolipid metabolism in the liver of obese and insulin-resistant Wistar rats fed with a high-fat high-sucrose diet as a representation of the Western feeding pattern. Despite a very limited amount of data, some studies indicated that an appropriate dose of CBG might alleviate liver damage secondary to non-alcoholic steatohepatitis (NASH) development in mice fed a methionine/choline-deficient (MCD) diet [30]. However, studies have been carried out on steatohepatitis mice models which do not exhibit insulin resistance, and our study aims to elucidate the influence of CBG in an animal model more similar to the Western diet observed in humans, which is the key pathogenetic factor of metabolic disorders [31]. Moreover, as it was shown in histological images, animals apart from obesity and insulin resistance, animals in this study also exhibited simple steatosis, a disease that is preliminary to NASH development. Thus, seeing a protective effect of CBG in this condition may give much more arsenal to treat all of the aforementioned disorders.

Encouraged by the previous results of our studies, which showed that chronic treatment with cannabidiol (CBD) during excessive fat intake led to an amelioration of imbalances in the sphingolipid pathway and improved insulin signaling in rat skeletal muscle, myocardium, brain, and adipose tissue, we wanted to extend our research and shed some light on another very poorly studied phytocannabinoid compound [28,32,33,34]. Thus, in our experiment, we formulated a hypothesis that CBG, by interfering with the sphingolipid metabolism pathway, may protect the liver from metabolic disturbances and the further development of diseases connected with it, such as IR. Our study indicated that treating insulin-resistant obese animals with CBG improved liver insulin sensitivity, which was seen in the increased phosphorylation of Akt in the phosphorylation sites of Ser and Thr, which increased Akt enzyme activity, and was also seen in the decreased phosphorylation of GSK-3β Tyr 216, which inactivated GSK enzyme activity. Although whole-body insulin resistance evaluated by ITT exhibited pronounced improvement after HFHS + CBG treatment only 30 min after insulin injection, calculated AUC-ITT did not show significant changes in the group treated with HFHS + CBG. These data may be explained by the fact that organ insulin resistance anticipates the occurrence of this metabolic disturbance in the whole organism. Moreover, as was found by the da Silva Rosa et al., studies on animals with knockout insulin signaling proteins showed that disruption in skeletal muscles, adipose tissue, and liver insulin signaling may cause more severe insulin resistance development with impaired body glucose homeostasis [35]. The main causes of such a situation are changes in lipid metabolism and the deposition of various lipid fractions such as sphingolipids. A similar conclusion was drawn by the Cinar et al., who found that in obese mice, intensified ceramide synthesis through the stimulation of the CB1 receptor in hepatocytes caused hepatic insulin resistance development [29,36].

In the sphingolipid metabolism pathway, we may distinguish the molecule that is a central hub in this pathway, namely ceramide (CER), and a plurality of its more complex derivatives. The primary pathway responsible for ceramide (CER) generation is the de novo synthesis pathway, in which serine and palmitoyl-CoA undergo condensation catalyzed by serine palmitoyltransferases (SPTLC1 and SPTLC2) [37]. In our study, we observed the lack of changes in the expression of ceramide de novo synthesis pathway enzymes after CBG treatment and the inhibition of this pathway after the HFHS diet. This is a particularly interesting phenomenon considering the high availability of substrates for de novo synthesis. Contrary to our results are findings from experiments conducted on skeletal muscles isolated from animals receiving high-fat diet and CBD injections. Scientists showed that CBD treatment significantly decreased ceramide deposition in high-fat-diet-fed animals through the inhibition of a de novo synthesis pathway [38]. As CBG is the direct precursor of CBD and the effect of both substances should be similar, the main difference between both studies was the type of diet that was used. Although the changes in SPTLC1 and SPTLC2 expression were not observed, the concentration of SFA, which is a precursor of CER in the de novo pathway, was affected. We observed a significant decrease in the hepatic concentration of SFA in the HFHS group, probably due to its transport of plasma or conversion to SFA1P by the SPHK1, after which increased expression was observed. However, in the HFHS + CBG group, the hepatic concentration of SFA increased compared to HFHS, raising the suspicion that its source may be also SFA1P, as this ‘side pathway’ may be bi-directional [25]. The existing scientific literature unequivocally demonstrates that an excess intake of foods abundant in saturated fatty acids amplifies the de novo ceramide synthesis pathway, leading to elevated CER levels in various tissues, including the liver. Interestingly, in our present study, the concentration of CER in the liver of rats fed a high-fat, high-sucrose diet, was almost intact, only significantly lowered in the group treated with CBG itself. Our results may be supported by the study by Taltavull et al., who also showed that the total ceramide content in the liver did not change under the influence of the HFHS diet [39]. However, there were changes in the qualitative composition of ceramides, as there was an increase in long-chain ceramides (LCcer), which are associated with IR, and a decrease in very long-chain ceramides (VLCcer), which are thought to prevent the development of IR [39]. Unfortunately, in our study, we did not assess the composition of ceramides after CBG treatment, which is considered to be the main weakness of our study. Importantly, the significantly higher plasma CER concentration was observed in the HFHS group, and was even more pronounced in response to HFHS + CBG treatment. It is most likely that the CER was formed in the liver but was exported into the plasma. We hypothesize that this may have been a compensatory mechanism of the liver to protect against the excessive deposition of CER, which has an effect on steatosis development. A similar conclusion was reached by Angrish et al. in their studies, who showed that in vivo lipid homeostasis is maintained through the activation of compensatory biological pathways which comprise fatty acid uptake, synthesis, efflux, and metabolism modification [40]. This may mean that under the influence of CBG, the removal of CER from the liver is facilitated, which would be a beneficial phenomenon, as this compound is considered harmful. In addition to de novo synthesis, CER may be produced from the degradation of sphingosine in the salvage pathway through the action of dihydroceramide synthase (CerS1-6) enzymes, which share these two routes of CER synthesis. In our study, we found that the expression of CerS5 was decreased in the HFHS + CBG group, and exactly the opposite change was observed in the expression of CerS6. Thus, in this case, we believe that the salvage pathway contributed little to overall sphingolipid metabolism. In addition, the lack of changes in tissue and plasma SFO levels, as well as alterations in CerS5 and CerS6 expressions, which overlapped, did not give any indication that this pathway was used. Contradictory data were presented in studies conducted by Berk et al. on rats receiving CBD injections after high-fat-diet-induced obesity where the salvage pathway in subcutaneous adipose tissue was inhibited. This may be explained by the fact that adipose tissue and the liver exhibit various alterations in lipid metabolism alterations in response to a diet rich in carbohydrates and lipids, as was the case in our study, or only in lipids, as in the study presented by Berk et al. [28]. 

The hydrolysis of sphingomyelin (SPH) is another pathway through which CER could be produced. However, sphingomyelin-to-ceramide conversion enzymes (Alk-Smase and N-SMase) contradict the fact that in our model sphingomyelin is a significant origin of ceramide. Furthermore, CER itself may be a source for SPH deposition. However, in our study the increased concentration of SPH in the liver altered by CBG administration in the HFHS group may be due to hyperplasia, because the main role of sphingomyelin is to be a part of the cell membrane. Another explanation for increased SPH accumulation could be the mitigation of damage to the liver from excess ceramide, because the conversion of CER back to SPH is less harmful to the liver. Once synthesized, CER can be degraded to sphingosine by ceramidases (ASAH 1-3), and sphingosine can then be converted to sphingosine-1-phosphate (S1P) by sphingosine kinases (SPHK1, SPHK2). In our study, we observed significant changes in the ceramide catabolism pathway. Specifically, in the HFHS + CBG group, we observed an increase in ceramide degradation, as the expression of the enzymes ASAH3 and SPHK1 increased significantly. These very similar changes are consistent with an increase in the liver concentration of S1P, the product of SPHK 1-2 action, confirming the augmentation of the catabolic pathway. Furthermore, we believe that there is no significant change in the concentration of SFO because it is only an intermediate product for the formation of S1P. Sphingosine-1-phosphate is one of several key molecules through which sphingolipids interact with the cell. However, it is important to note that S1P can only exert its effects after binding to specific extracellular sphingosine-1-phosphate receptors (S1PR), mediating downstream signaling cascades and the subsequent regulation of diverse physiological functions such as cell proliferation, migration, and immune responses [41]. In the liver, these receptors are mainly S1PR2 and S1PR3 [42]. Our study showed that most of the formed S1P are retained in the liver, as we observed reduced plasma S1P concentrations, which is consistent with the low expression of the S1P receptor namely S1PR2 in HFHS + CBG, which means reduced binding of S1P to this receptor. Another important factor regulating the impact of S1P on cells is the regulation of its quantity through transport. The export of S1P out of the cell involves different mechanisms. The first one involves sphingosine phosphate-specific phosphatase 2 (SPSN2), which dephosphorylates S1P, converting it back into sphingosine for extracellular release. Another mechanism involves ATP-binding cassette transporter A1 (ABCA1), a transmembrane protein that facilitates the efflux of S1P from the cell to the extracellular environment. Both SPSN2 and ABCA1 contribute to the regulation of S1P levels in the liver cells [43]. In our study, the expression of ABCA1 significantly declined in the HFHS + CBG group, which confirms the accumulation of S1P in the cell rather than their export. On the other hand, within the cell, S1P is transported by a protein called ceramide transfer protein (CERT). CERT plays a crucial role in transferring S1P from the endoplasmic reticulum (ER) to the Golgi apparatus, where S1P is further processed to other sphingolipids such as sphingomyelin. We observed the down-regulation of CERT expression in the HFHS group, but CBG did not change it significantly. Thus, we may assume that this pathway was not the main one responsible for increased SPH deposition. The last scenario for S1P is the irretrievable degradation to phosphoethanolamine and an unsaturated fatty aldehyde hexadecanol, through the action of Sphingosine-1-Phosphate Lyase 1 (SGPL1). The degradation of S1P contributes to the turnover of sphingolipids and regulates S1P signaling by maintaining appropriate levels of S1P within the cell [25]. In our study, the decomposition of S1P was not a significant factor influencing S1P content in the liver, as we did not observe any change in SGLP1 expression in the HFHS + CBG group, supporting the idea that synthesized S1P accumulates in the cell. The data demonstrate that S1P accumulation within liver cells can multi-directionally influence their homeostasis. Circulating S1P bound to apolipoprotein M (ApoM), a component of high-density lipoprotein (HDL), can protect the liver of C57BL/6 mice from LPS-induced apoptosis and inflammatory injuries [44]. Furthermore, S1P synthesis in hepatocytes after the activation of S1PR1 in liver sinusoidal endothelial cells has been shown to induce cell proliferation and thus promote liver regeneration after ischemic/reperfusion injury or partial hepatectomy in rats [45]. What is more, S1P can protect sinusoidal endothelial cells in rat liver from apoptosis in response to ethanol [46] and rat primary hepatocytes from apoptosis due to tumor necrosis factor-alpha [47]. However, the role of S1P in the liver is not so clear, as the S1P signaling pathway has also been shown to be involved in liver inflammation and fibrosis [41]. In the human liver, the expression of SPHK1 (S1P-synthesizing enzyme) and S1PR2 (S1P receptor) correlates with the severity of liver fibrosis, while, interestingly, the levels of S1P itself and the expression of enzymes responsible for its breakdown were not increased [48]. In addition, there is evidence from one study that the inhibition of SPHK1, an enzyme that generates S1P from SFO, protects hepatocytes from apoptosis [49]. Although various effects of S1P in the liver have been partially established, its exact role, in particular its influence on the development of liver steatosis and fibrosis, is not yet fully understood. Thus, taking into account these contradictory data, we could not determine whether S1P deposition after CBG treatment in the HFHS group is beneficial or not. We do believe that another study with the use of SPHK inhibition should be conducted to elucidate this issue.

## 5. Conclusions

Our study aimed to investigate the effect of cannabigerol (CBG) on the liver sphingolipid metabolism pathway. We observed that CBG administration in a rat model of insulin resistance induced by a high-fat high-sucrose diet had significant effects on the sphingolipid metabolism pathway in the liver. Our findings suggest that CBG treatment modulates the sphingolipid metabolism pathway by influencing the synthesis, degradation, and transport of sphingolipids in the liver. Specifically, CBG administration led to the intensified export of ceramide to the plasma, potentially preventing the development of liver steatosis. Furthermore, CBG treatment resulted in the increased degradation of ceramides into sphingosine and enhanced the conversion of sphingosine to S1P, which may have a positive impact on insulin sensitivity. However, our study had some limitations. We did not assess the composition of ceramides, which could provide further insights into the specific changes induced by CBG treatment. Additionally, we did not examine the downstream signaling cascades and physiological functions mediated by S1P binding to its receptors in the liver. Most importantly, our study lacked direct evidence that CBG improved insulin resistance through changes in the sphingolipid accumulation. To fill this gap in knowledge, some experiments with the use of inhibitors from the ceramide synthesis pathway, e.g., Myriocin, should be conducted. Such data could shed light on the understanding of the CBGs’ mechanism of action if it has a direct influence on insulin signaling proteins or induces changes in sphingolipid content in the liver. Despite these limitations, our study highlights the potential of CBG as a novel therapeutic approach for insulin resistance and metabolic disorders associated with impaired sphingolipid metabolism. Further research is warranted to fully understand the mechanisms underlying the effects of CBG and to explore its clinical implications in the management of insulin resistance and related metabolic conditions.

## Figures and Tables

**Figure 1 nutrients-15-04350-f001:**
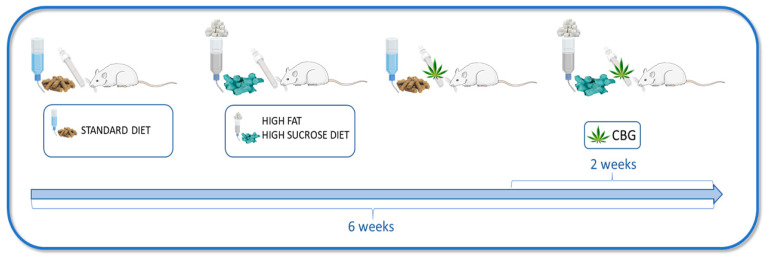
Schematic presentation of the experimental design. The Figure was created with the use of Server Medical Art.

**Figure 2 nutrients-15-04350-f002:**
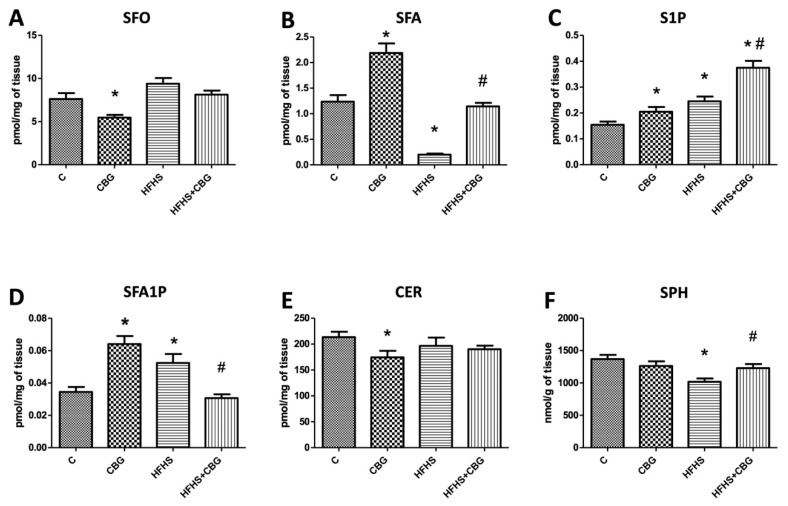
Concentration of sphingosine (SFO) (**A**), sphinganine (SFA) (**B**), sphingosine-1-phosphate (S1P) (**C**), sphinganine-1-phosphate (SFA1P) (**D**), ceramide (CER) (**E**), and sphingomyelin (SPH) (**F**) in rat’s liver. The quantification of sphingomyelin concentration was conducted using gas–liquid chromatography, while the measurement of other sphingolipids was performed via high-performance liquid chromatography, following the procedures detailed in the Section 2. The results are presented as mean ± SD and the data are from n = 10 rats; *—*p* < 0.05 significant difference vs. control group, #—*p* < 0.05 significant difference vs. HFHS group.

**Figure 3 nutrients-15-04350-f003:**
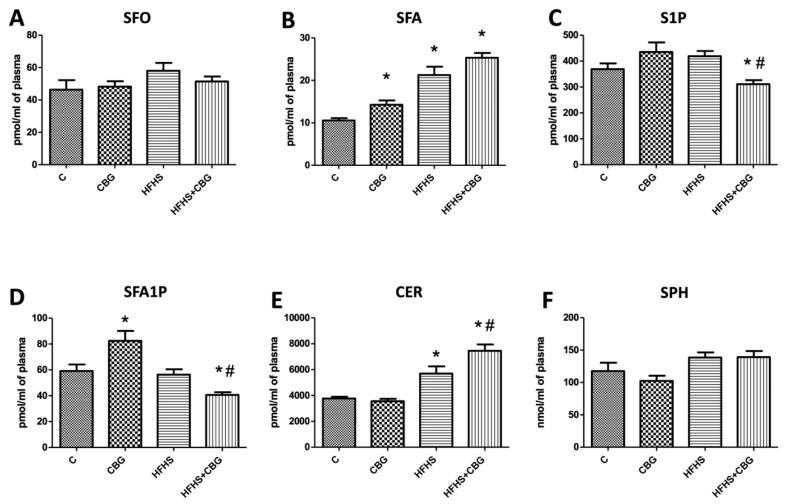
Concentration of sphingosine (SFO) (**A**), sphinganine (SFA) (**B**), sphingosine-1-phosphate (S1P) (**C**), sphinganine-1-phosphate (SFA1P) (**D**), ceramide (CER) (**E**), and sphingomyelin (SPH) (**F**) in rat’s plasma. The quantification of sphingomyelin concentration was conducted using gas–liquid chromatography, while the measurement of other sphingolipids was performed via high-performance liquid chromatography, following the procedures detailed in the Section 2. The results are presented as mean ± SD, and the data are from n = 10 rats; *—*p* < 0.05 significant difference vs. control group, #—*p* < 0.05 significant difference vs. HFHS group.

**Figure 4 nutrients-15-04350-f004:**
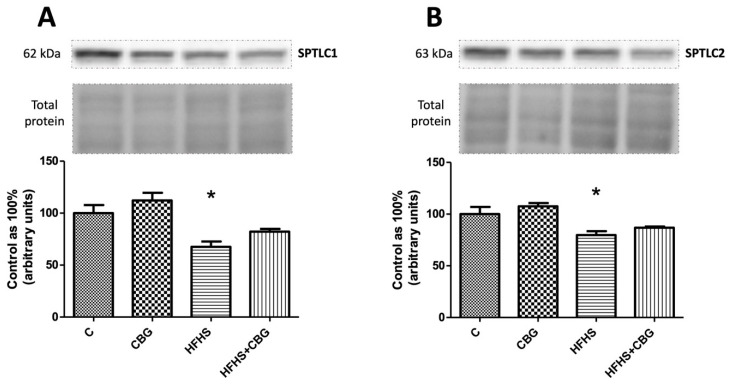
The expression of serine palmitoyltransferase 1 (SPTLC1) (**A**) and serine palmitoyltransferase 2 (SPTLC2) (**B**) in rat’s liver and the images of representative blots of target protein and corresponding total protein. The evaluation of enzymes specific to the de novo ceramide synthesis pathway was conducted through Western blotting, following the protocols outlined in the Section 2. The results are presented as mean ± SD, and the data are from n = 6 rats; *—*p* < 0.05 significant difference vs. control group.

**Figure 5 nutrients-15-04350-f005:**
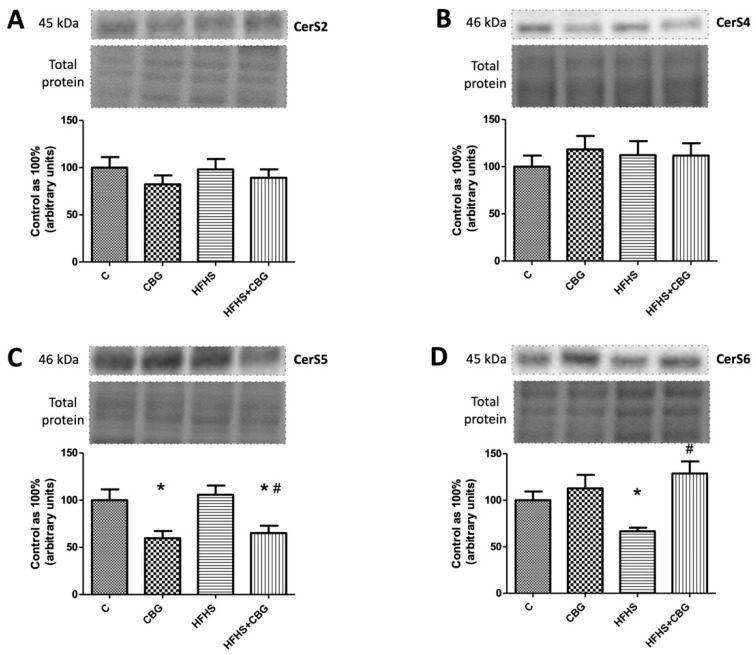
The expression of dihydroceramide synthase 2 (CerS2) (**A**), dihydroceramide synthase 4 (CerS4) (**B**), dihydroceramide synthase 5 (CerS5) (**C**), and dihydroceramide synthase 6 (CerS6) (**D**) in rat’s liver, as well as the images of representative blots of target protein and corresponding total protein. The expression of enzymes that share salvage and de novo ceramide synthesis pathways was assessed by Western blotting, as described in the Section 2. The results are presented as mean ± SD and the data are from n = 6 rats; *—*p* < 0.05 significant difference vs. control group, #—*p* < 0.05 significant difference vs. HFHS group.

**Figure 6 nutrients-15-04350-f006:**
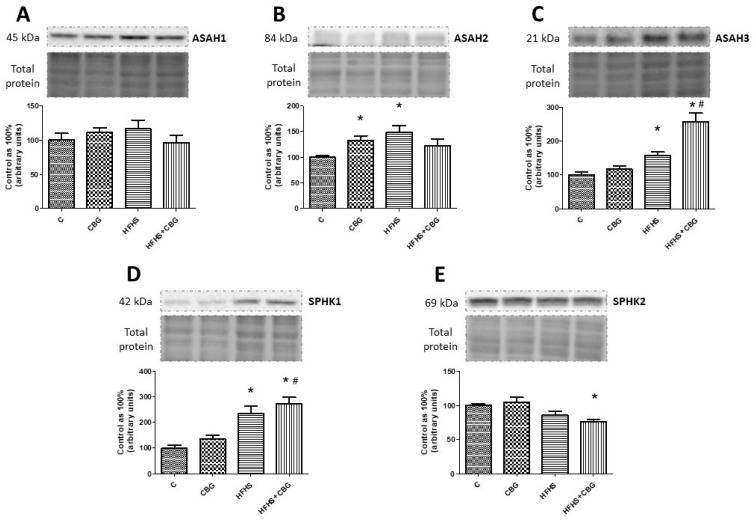
The expression of acid ceramidase (ASAH1) (**A**), neutral ceramidase (ASAH2) (**B**), alkaline ceramidase (ASAH3) (**C**), sphingosine kinase 1 (SPHK1) (**D**), and sphingosine kinase 2 (SPHK2) (**E**) in rat’s liver, as well as the images of representative blots of target protein and corresponding total protein. The expression of enzymes from the ceramide catabolic pathway was assessed by Western blotting, as described in the Section 2. The results are presented as mean ± SD and the data are from n = 6 rats; *—*p* < 0.05 significant difference vs. control group, #—*p* < 0.05 significant difference vs. HFHS group.

**Figure 7 nutrients-15-04350-f007:**
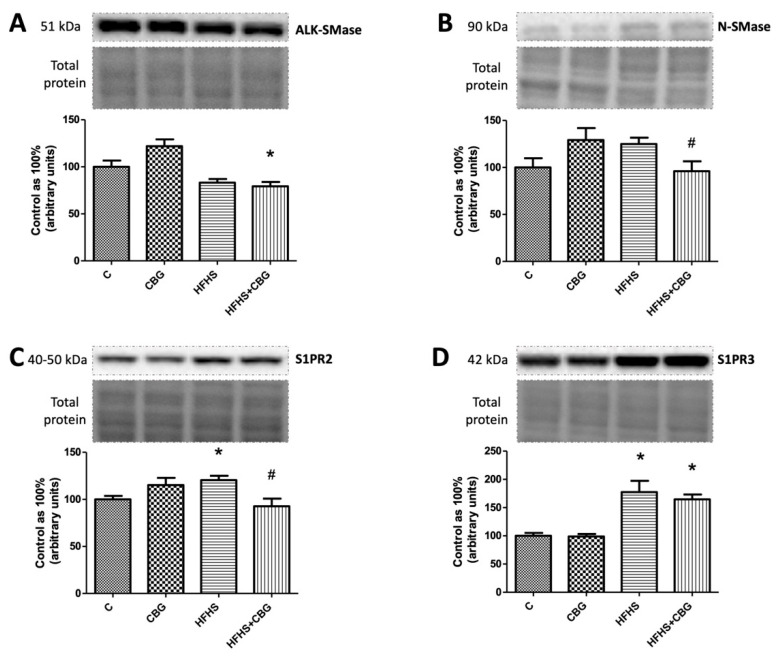
The expression of alkaline sphingomyelinase (AlK-SMase) (**A**), neutral sphingomyelinase (N-SMase) (**B**), sphingosine-1-phosphate receptor 2 (S1PR2) (**C**), and sphingosine-1-phosphate receptor 3 (S1PR3) (**D**) in rat’s liver, as well as the images of representative blots of target protein and corresponding total protein. The expression of enzymes from hydrolysis of the sphingomyelin pathway and receptors for S1P was assessed by Western blotting, as described in the Section 2. The results are presented as mean ± SD, and the data are from n = 6 rats; *—*p* < 0.05 significant difference vs. control group, #—*p* < 0.05 significant difference vs. HFHS group.

**Figure 8 nutrients-15-04350-f008:**
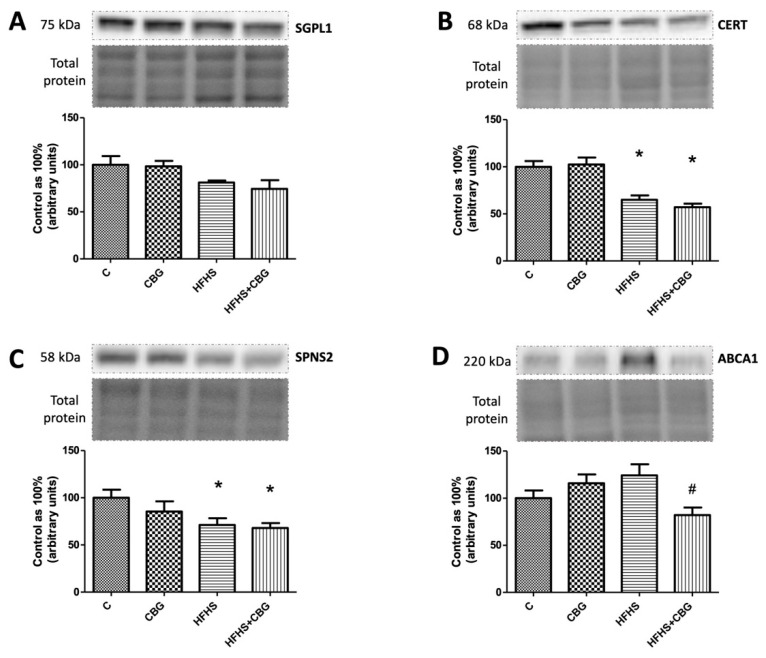
The expression of sphingosine-1-phosphate lyase 1 (SGPL1) (**A**), ceramide transport protein (CERT) (**B**), sphingolipid transporter 2 (SPNS2) (**C**), and ATP-binding cassette transporter (ABCA1) (**D**) in rat’s liver, as well as the images of representative blots of the target protein and corresponding total protein. The expression of S1P-degrading enzyme and sphingolipid transporting proteins was assessed by Western blotting, as described in the Section 2. The results are presented as mean ± SD and the data are from n = 6 rats; *—*p* < 0.05 significant difference vs. control group, #—*p* < 0.05 significant difference vs. HFHS group.

**Figure 9 nutrients-15-04350-f009:**
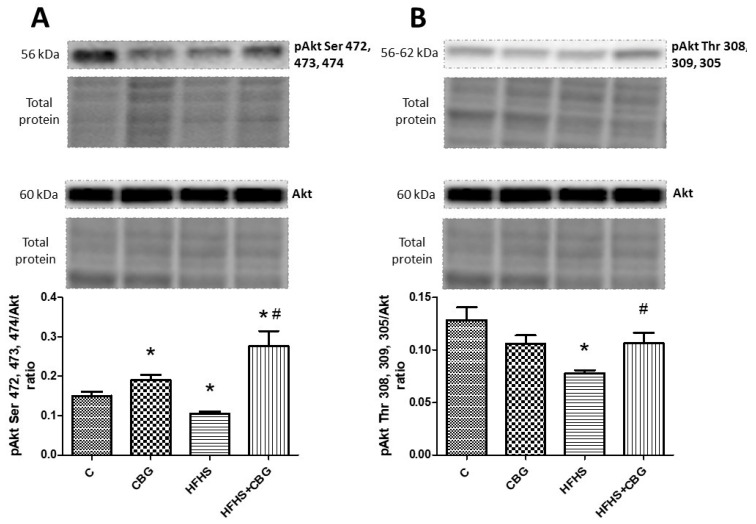
The expression ratio of phosphorylated protein kinase B in Ser 472, 473, and 474 (pAkt Ser 472, 473, and 474) to total protein kinase B (Akt) (**A**), and phosphorylated protein kinase B in Thr 308, 309, and 305 (pAkt Thr 308, 309, and 305) to total protein kinase B (Akt) (**B**) in rat’s liver, as well as the images of representative blots of target protein and corresponding total protein. The assessment of phosphorylated and total proteins pertaining to the insulin signaling pathway was conducted through Western blotting, following the procedures elucidated in the Section 2. The results are presented as mean ± SD, and the data are from n = 6 rats; *—*p* < 0.05 significant difference vs. control group, #—*p* < 0.05 significant difference vs. HFHS group.

**Figure 10 nutrients-15-04350-f010:**
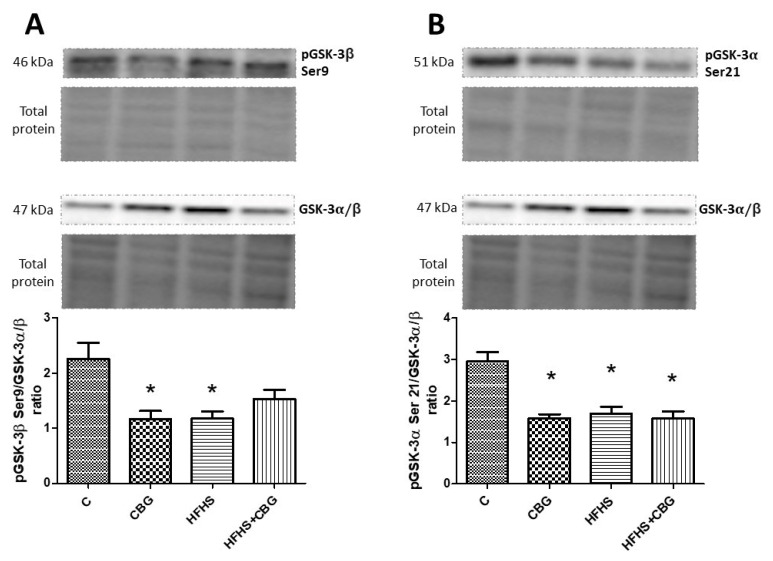
The expression ratio of phosphorylated glycogen synthase kinase B-3β Ser 9 (pGSK-3β Ser 9) to total glycogen synthase kinase B-3alpha/beta (GSK-3α/β) (**A**), phosphorylated glycogen synthase kinase B-3α Ser 21 (pGSK-3α Ser 21) to total glycogen synthase kinase B-3α/β (GSK-3α/β) (**B**) in rat’s liver, as well as the images of representative blots of target protein and corresponding total protein. The assessment of phosphorylated and total proteins pertaining to the insulin signaling pathway was conducted through Western blotting, following the procedures elucidated in the Section 2. The results are presented as mean ± SD, and the data are from n = 6 rats; *—*p* < 0.05 significant difference vs. control group.

**Figure 11 nutrients-15-04350-f011:**
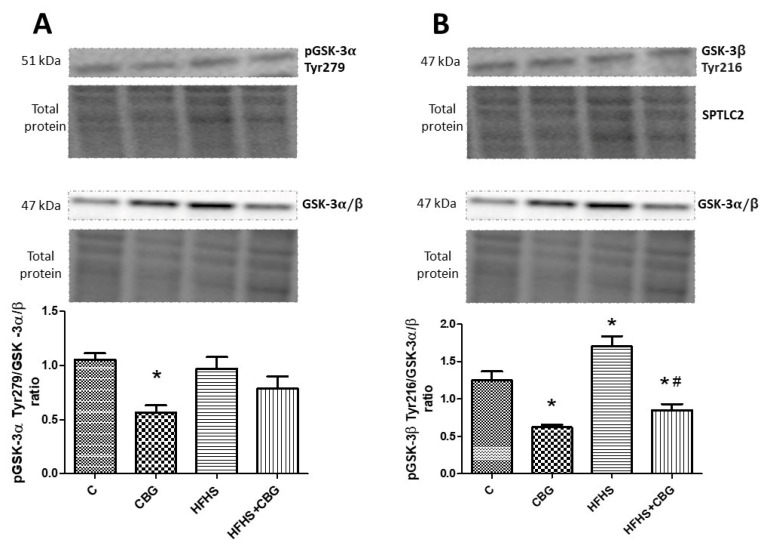
The expression ratio of phosphorylated glycogen synthase kinase B-3α Tyr 279 (pGSK-3α Tyr 279) to total glycogen synthase kinase B-3α/β (GSK-3α/β) (**A**) and phosphorylated glycogen synthase kinase B-3β Tyr 216 (pGSK-3β Tyr 216) to total glycogen synthase kinase B-3α/β (GSK-3α/β) (**B**) in rat’s liver, as well as the images of representative blots of target protein and corresponding total protein. The assessment of phosphorylated and total proteins pertaining to the insulin signaling pathway was conducted through Western blotting, following the procedures elucidated in the Section 2. The results are presented as mean ± SD and the data are from n = 6 rats; *—*p* < 0.05 significant difference vs. control group, #—*p* < 0.05 significant difference vs. HFHS group.

**Figure 12 nutrients-15-04350-f012:**
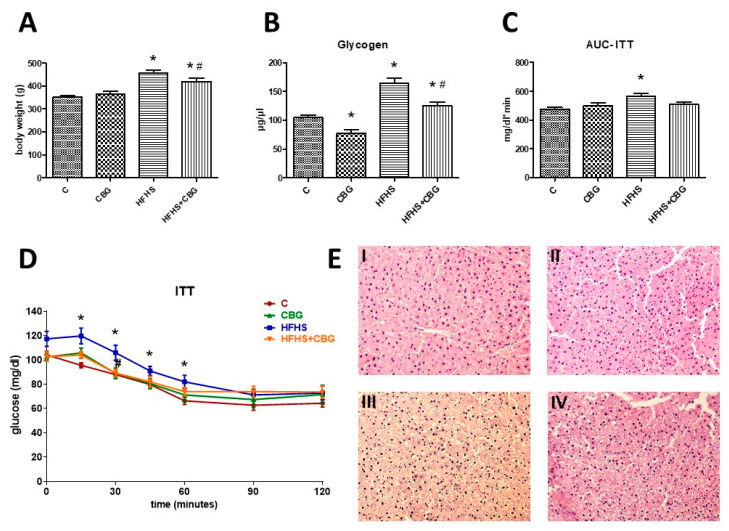
The body weight of rats at the end of the experiment (**A**), liver glycogen concentration (**B**), the area under the curve during the Insulin Tolerance Test—ITT (AUC-ITT) (**C**), the blood glucose levels during the ITT (**D**), as well as representative histological images (**E**) showing hematoxylin–eosin (H + E) staining of the hepatic tissue from the control group (I), CBG group (II), HFHS group (III), and HFHS + CBG group (IV). The blood glucose levels were assessed using a glucose analyzer and test strips, and glycogen concentration was determined using a commercially available kit, as described in the Materials and Methods Section 2. The results are presented as mean ± SD and the data are from n = 6 rats (ITT and AUC-ITT) or n = 10 rats (body weight and glycogen level); *—*p* < 0.05 significant difference vs. control group, #—*p* < 0.05 significant difference vs. HFHS group.

**Figure 13 nutrients-15-04350-f013:**
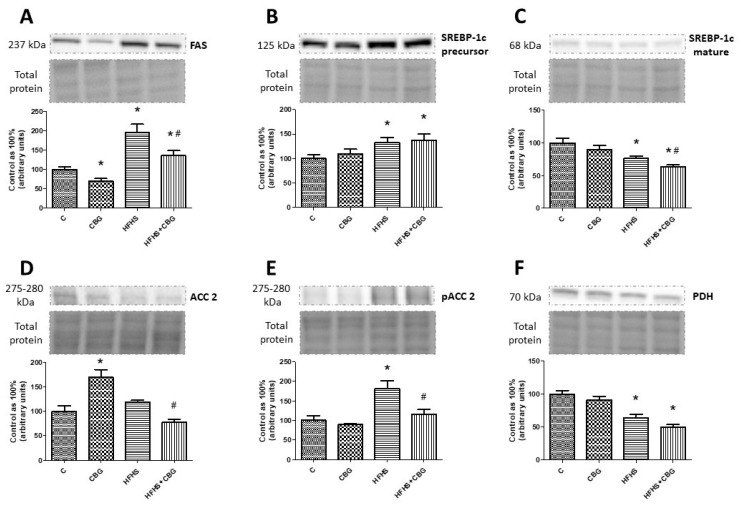
The expression of fatty acid synthase (FAS) (**A**), sterol regulatory element-binding protein-1c precursor (SREBP-1c) (**B**), SREBP-1c mature (**C**), acetyl-CoA carboxylase 2 (ACC2) (**D**), phosphorylated acetyl-CoA carboxylase 2 Ser 9 (pACC2 Ser 9) (**E**), and pyruvate dehydrogenase (PDH) (**F**) in rat’s liver, as well as the images of representative blots of target protein and corresponding total protein. The expression of the above proteins was assessed using Western blotting, as described in the Section 2. The results are presented as mean ± SD and the data are from n = 6 rats; *—*p* < 0.05 significant difference vs. control group, #—*p* < 0.05 significant difference vs. HFHS group.

## Data Availability

The data presented in this study are available on request from the corresponding author.

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
