# Peer review of "How Does CBG Administration Affect Sphingolipid Deposition in the Liver of Insulin-Resistant Rats?"

_nutrients, 2023, doi:10.3390/nu15204350_

Round 1

Reviewer 1 Report

How does CBG administration affect sphingolipid deposition in the liver of insulin-resistant rats? 

Wiktor Bzdęga et al.

Summary: The authors investigated the effect of cannabigerol (CBG) on insulin resistance in rats fed a high fat/high carbohydrate diet. The specific focus of the study was on sphingolipid metabolism. CBG did not elevate ceramide synthesis, but the metabolism of ceramide was accelerated. Improvements in insulin resistance were observed with CBG administration.

Comments:

The strength of this manuscript is in the assessment of sphingosine/ceramides, its metabolites and the enzymes involved in sphingosine metabolism in CBG treated rats. The manuscript is well-written, and the data are clear. The main concern of this reviewer is whether CBG affects insulin resistance via changes in ceramide. The link between sphingosine metabolism and insulin resistance is not well developed. It is not clear if the effects on the ITT (shown in Fig 12A) are due to direct actions of CBG or changes in sphingosine metabolites.

1. The authors should comment on the effects of the HFHS diet and CBG. Some concerns would include: Did CBG affect the food intake? Did the HFHS diet rats have a fatty liver? Were measurements of body lipid mass made? Were insulin levels measured? The changes in body weight were assessed in figure 12C.

2. Regarding figure 2 and 3, were the rats fed or fasted before the sphingolipid measurements?

3. Regarding changes in ceramide in the plasma (fig 3), is the source of ceramide only from the liver or are multiple tissues involved? How would this affect the results.

4. Did the authors examine markers of hepatic lipogenesis or gluconeogenesis such as nSREBP-1c, FAS, ACC or PEPCK at the mRNA or protein level? These would be considered markers of insulin resistance. 

5. The studies of Akt and GSK-3 phosphorylation (figs 9,10,11) were good. It would help the reader to mention in the text whether the phosphorylation increases or decreases enzyme activity and the relation to insulin resistance.

6. Has any work been done on CBG and its effect in isolated hepatocytes with respect to metabolic pathways? If so, it would be good to add to the discussion. The question remains as to whether the metabolic effects of CBG are direct or through ceramides.

Reviewer 2 Report

There is enough published data along with reports showing that Cannabidiol helps in Metabolic Syndrome along with mechanistic studies. Given the already available information, it is pertinent that authors should have compared their results with other Cannabidiol besides using CBG. This report is insufficient in the current form.

Please see the following review article for more info:

Wiciński M, Fajkiel-Madajczyk A, Kurant Z, Gryczka K, Kurant D, Szambelan M, Malinowski B, Falkowski M, Zabrzyński J, Słupski M. The Use of Cannabidiol in Metabolic Syndrome-An Opportunity to Improve the Patient's Health or Much Ado about Nothing? J Clin Med. 2023 Jul 11;12(14):4620. doi: 10.3390/jcm12144620. PMID: 37510734; PMCID: PMC10380672.

NA
